# Practical Hybrid Quantum Language Models with Observable Readout on Real Hardware

## Abstract

Hybrid quantum-classical models are emerging as a key approach for leveraging near-term quantum devices. We present quantum recurrent neural networks (QRNNs) and quantum convolutional neural networks (QCNNs) as hybrid quantum language models, and demonstrate for the first time generative language modeling trained and evaluated on real quantum hardware. Our models combine parametric quantum circuits with a lightweight classical projection layer, using hardware-friendly multi-sample SPSA to train the quantum parameters efficiently, and standard gradient-based updates for the classical weights. To support evaluation, we construct and release a synthetic dataset for next-word prediction. Experiments on both sentence classification and language modeling tasks show that QRNNs and QCNNs can be trained end-to-end on NISQ devices and achieve competitive performance in low-resource regimes. These results establish quantum sequence models as a promising foundation for quantum natural language processing.

## 1 Introduction

Natural language processing (NLP) has seen remarkable advances in recent years, largely driven by deep learning architectures such as recurrent neural networks (RNNs) (Elman, 1990; Hochreiter & Schmidhuber, 1997), convolutional neural networks (CNNs) (Kim, 2014), and Transformers (Vaswani et al., 2017). These models have enabled impressive performance across a range of tasks, including language modeling, machine translation, and text generation. However, scaling these architectures to handle large vocabularies, long sequences, or limited data regimes can be computationally expensive and data-hungry, motivating the exploration of alternative computational paradigms.

Quantum computing has recently emerged as a promising candidate to enhance machine learning algorithms by leveraging the principles of superposition, entanglement, and interference (Nielsen & Chuang, 2010; Schuld et al., 2015). In particular, hybrid quantum-classical models, where parametric quantum circuits are combined with classical post-processing layers, have attracted significant attention due to their compatibility with noisy intermediate-scale quantum (NISQ) devices (Preskill, 2018; Benedetti et al., 2019). These models offer the potential for richer function representation and novel inductive biases while remaining trainable using classical optimization strategies.

While a variety of quantum approaches to NLP have been proposed, most works remain largely theoretical and validate their methods only on simulators. The DisCoCat framework (Coecke et al., 2010) pioneered a categorical connection between compositional semantics and quantum circuits, inspiring several follow-up studies on quantum natural language processing (Meichanetzidis et al., 2020; Blacoe et al., 2013). However, these efforts typically emphasize conceptual formalisms or small-scale simulation results. A notable exception is the work of Lorenz et al. (2023), who demonstrated sentence classification using the DisCoCat framework on real quantum hardware, establishing the first experimental evidence that QNLP can be made practical. Despite these advances, systematic investigations of quantum sequence models—such as quantum recurrent and convolutional architectures—remain scarce, particularly for generative tasks like language modeling.

Our work addresses this gap by developing QRNNs and QCNNs that can be trained and evaluated end-to-end on NISQ devices, moving beyond proof-of-principle demonstrations toward more general-purpose quantum NLP models. Our hybrid quantum language models (HQLMs) combine parametric quantum circuits with a lightweight classical projection layer, using hardware-friendly multi-sample SPSA to train quantum parameters and gradient-based updates for classical weights. We evaluate

Figure 1: Overview of our hybrid quantum language models (HQLMs). Tokens are embedded into quantum states, processed by QRNN or QCNN layers, and mapped to predictions via a classical projection head, trained end-to-end with multi-sample SPSA and gradient-based updates.

these architectures on both simulated environments and real quantum hardware, demonstrating feasibility in practice and competitive performance on both synthetic and natural language datasets, including next-word prediction and classification tasks.

**Contributions**   Our work makes the following contributions:

1. We propose **hybrid quantum language models** based on QRNNs and QCNNs, designed for sequence modeling tasks in NLP.
2. We introduce a **scalable training framework** using multi-sample SPSA for quantum parameters and gradient-based updates for classical layers, enabling end-to-end training on NISQ devices.
3. We analyze the role of **quantum embeddings**, Z and ZZ observable-based feature extraction, and architectural trade-offs, including circuit depth, number of qubits, and shot noise.
4. We provide an **empirical evaluation** on synthetic natural language datasets, showing competitive performance with classical baselines in low-resource regimes and robustness to quantum noise.
5. We report the first **set of experiments** training these hybrid language models on real quantum hardware for generative language modeling, establishing practical feasibility.

Our results suggest that hybrid quantum architectures are a viable direction for enhancing NLP pipelines, offering new avenues for efficient and expressive sequence modeling in the NISQ era.

## 2 BACKGROUND AND NOTATION

We briefly introduce the concepts underlying our hybrid quantum language models (HQLMs). Notation details are provided in App. B.

### 2.1 LANGUAGE MODELING

Language modeling estimates the probability distribution of token sequences. Given a sequence of tokens $(x_1, \ldots, x_T)$, a model learns to predict the probability

$$P(x_1, \ldots, x_T) = \prod_{t=1}^{T} P(x_t \mid x_{<t}), \tag{1}$$

typically in an autoregressive fashion. The objective is to capture syntactic and semantic dependencies so that the model can predict the next token from context. Evaluation metrics such as cross-entropy loss or perplexity measure alignment with observed sequences.

### 2.2 CLASSICAL NEURAL ARCHITECTURES

Classical neural language models map token embeddings to next-token probabilities using architectures that model sequential structure. *Recurrent networks* (RNNs, LSTMs, GRUs) process tokens step-by-step via hidden states. *Convolutional networks* (CNNs) apply 1D convolutions to capture local n-gram patterns, extended with dilations for longer contexts. *Transformers* replace recurrence with attention, modeling pairwise dependencies across the sequence.

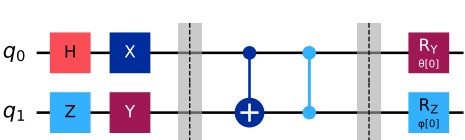

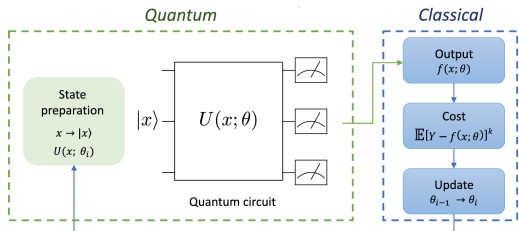

Figure 2: Examples of elementary quantum gates, the building blocks of quantum circuits: single-qubit gates ($H$, $X$, $Y$, $Z$), two-qubit entangling gates (CNOT, CZ), and single-qubit parametric gates ($R_Y(\theta)$, $R_Z(\phi)$).

Figure 3: Schematic of a Variational Quantum Algorithm (VQA) workflow from Macaluso et al. (2020): a parametric quantum circuit (PQC) is optimized by a classical iterative algorithm.

### 2.3 QUANTUM COMPUTING BASICS

Quantum computation is based on *qubits*, which can exist in superpositions such as

$$|\psi\rangle = \alpha\,|0\rangle + \beta\,|1\rangle, \quad |\alpha|^2 + |\beta|^2 = 1. \tag{2}$$

Multiple qubits form states in tensor-product spaces, enabling *entanglement*, i.e., correlations without classical analogues. Quantum gates are unitary operators; standard examples include the Hadamard ($H$), Pauli-$X$, and the entangling CNOT. Figure 2 shows the elementary building blocks, including parametrized single-qubit rotations $R_Y(\theta)$ and $R_Z(\phi)$.

### 2.4 PARAMETERIZED QUANTUM CIRCUITS (PQCS)

Parameterized quantum circuits (PQCs), also known as variational circuits, are the quantum analogue of neural networks. They consist of input encodings, layers of parametrized gates $U(\boldsymbol{\theta})$, and measurement. Given a classical input $\boldsymbol{x}$, the circuit produces expectation values

$$f_{\boldsymbol{\theta}}(\boldsymbol{x}) = \langle\phi(\boldsymbol{x})|\,U^{\dagger}(\boldsymbol{\theta})OU(\boldsymbol{\theta})\,|\phi(\boldsymbol{x})\rangle, \tag{3}$$

where $O$ is an observable (e.g., $Z$ or $ZZ$ operators). The parameters $\boldsymbol{\theta}$ are trained in a hybrid loop with a classical optimizer. An overview of this framework is illustrated in Figure 3.

### 2.5 READOUT ON NISQ DEVICES

Noisy intermediate-scale quantum (NISQ) devices are limited by circuit depth and gate errors, making PQC design a trade-off between expressivity and noise resilience. A key choice is how to extract classical features from the quantum state.

The *sampling approach* measures computational basis states directly, interpreting outcomes as token probabilities. While intuitive, it often complicates optimization, since probability mass must align with discrete encodings.

The alternative, used here, is *estimator-based readout*: computing expectation values of observables to obtain continuous feature vectors. These features are then mapped by a classical linear layer into token logits: $\ell = W\boldsymbol{f} + \boldsymbol{b}$, with $W \in \mathbb{R}^{d \times V}$. This hybrid readout smooths optimization and integrates naturally with classical training while remaining feasible on real hardware.

## 3 RELATED WORK

### 3.1 QUANTUM MACHINE LEARNING (QML)

Quantum machine learning studies how quantum circuits can augment or replace elements of classical learning algorithms (Biamonte et al., 2017; Schuld et al., 2015). A central paradigm is the use of variational quantum circuits (VQCs), where parametrized gates define expressive models trained with classical optimization (McClean et al., 2016; Schuld et al., 2020; Benedetti et al., 2019). These hybrid quantum-classical approaches are well-suited for noisy intermediate-scale quantum (NISQ) devices (Preskill, 2018), as they exploit quantum superposition and entanglement while relying on lightweight classical layers for readout and stability.

VQCs face distinctive challenges, including barren plateaus (McClean et al., 2018), which motivate research into circuit expressibility and entangling capacity (Sim et al., 2019), as well as mitigation

strategies such as architectural constraints or residual connections (Kashif & Al-Kuwari, 2024). Beyond classification and regression, quantum circuits have been extended to recurrent models (Bausch, 2020; Macaluso et al., 2020), convolutional designs (Cong et al., 2019), and natural language tasks (Coecke et al., 2010; Meichanetzidis et al., 2020; Lorenz et al., 2023). Surveys provide broader overviews of algorithms and applications (Jerbi et al., 2023; Chen et al., 2024; Nausheen et al., 2025). Nevertheless, most work remains simulator-based or limited to small-scale proofs of concept, leaving open the question of whether hybrid quantum models can be trained end-to-end on real hardware.

### 3.2 QUANTUM NATURAL LANGUAGE PROCESSING (QNLP)

Quantum NLP builds on the compositional distributional (DisCoCat) framework (Coecke et al., 2010), which maps grammatical structure to tensor networks and, in turn, quantum circuits. Early works explored variational circuits for question answering and entailment (Lorenz et al., 2023; Meichanetzidis et al., 2020), quantum embeddings for token representations (Panahi et al., 2019; Chen et al., 2021), and bag-of-words style models (Lorenz et al., 2023). While these approaches demonstrate feasibility, they are typically limited to shallow circuits, small-scale tasks, or simulators, and do not address full sequence modeling.

In parallel, recent studies investigate quantum adaptations of Transformer architectures, including residual designs and attention mechanisms (Khatri et al., 2024; Liao & Ferrie, 2024; Amire, 2025; Tomal et al., 2025). Although promising, these models often require deeper circuits and more qubits, making them challenging to deploy on current NISQ devices. In contrast, we focus on QRNNs and QCNNs, which provide a more hardware-efficient approach to sequential quantum processing. Our work complements prior efforts by presenting the first end-to-end training and evaluation of hybrid quantum sequence models for generative language modeling on real quantum hardware.

### 3.3 QUANTUM RECURRENT NEURAL NETWORKS (QRNNs)

QRNNs implement sequential processing through repeated application of parametrized unitaries acting on two registers: a short-lived *embedding* register (encoding the current token) and a longer-lived *hidden* register carrying memory across time steps (Bausch, 2020; Meichanetzidis et al., 2020). Each step prepares an embedding state, entangles it with the hidden register via a recurrent block, and updates the hidden state for the next step. Optimization typically combines gradient-free methods or parameter-shift rules with classical output layers (Schuld et al., 2020; Jerbi et al., 2023). Challenges include residual entanglement between registers, accumulation of hardware noise in long unrollings, and choices of readout observables (Widdows et al., 2024b). Despite these limitations, QRNNs provide a natural choice for sequence models and remain a key candidate for quantum NLP.

### 3.4 QUANTUM CONVOLUTIONAL NEURAL NETWORKS (QCNNs)

QCNNs generalize classical convolutional and pooling operations to quantum circuits, using local entangling unitaries and qubit reduction to extract hierarchical features (Cong et al., 2019; Hur et al., 2022). While most prior applications focus on classical data classification, QCNNs have also been adapted for token-level NLP tasks (Meichanetzidis et al., 2020; Widdows et al., 2024b). Their parallel structure reduces circuit depth compared to QRNNs, making them a practical choice for NISQ devices, though trade-offs remain between expressivity and hardware feasibility.

## 4 METHOD

We introduce hybrid quantum language models (HQLMs) that adapt recurrent and convolutional neural architectures to parameterized quantum circuits (PQCs).

### 4.1 TOKEN EMBEDDINGS

We adopt $R_y$ angle embeddings as our input encoding. Each token $t \in V$ has a trainable vector $\boldsymbol{\theta}_t \in \mathbb{R}^d$, mapped to a separable quantum state

$$|\psi_t\rangle = \bigotimes_{j=1}^{d} R_y(\theta_{t,j}) |0\rangle. \tag{4}$$

This scheme is shallow, noise-robust, and hardware-friendly, as it avoids entangling gates and allows virtual qubits to be distributed across non-adjacent physical qubits. While richer encodings (e.g. amplitude or entangled maps (Havlíček et al., 2019; Schuld & Killoran, 2019; Pérez-Salinas et al., 2020; Lloyd et al., 2020)) exist, they incur greater circuit depth and complexity, leading to increased noise and decoherence. We leave exploration of such embeddings to future work.

## 4.2 PQC LAYERS AS NEURAL BLOCKS

Following prior QML and QNLP work (Sim et al., 2019; Schuld et al., 2020; Bausch, 2020; Meichanetzidis et al., 2020), we use shallow PQCs as quantum analogues of neural layers. Each layer applies parameterized rotations on each qubit ($R_y$, $R_z$) interleaved with an entangling layer of pairwise CNOTs. This balances expressivity and noise resilience (Sim et al., 2019; Schuld & Petruccione, 2021), crucial for NISQ devices where deep entangling networks quickly decohere (Preskill, 2018). For more expressivity and entangling power, we can stack multiple layers. We use these PQCs as recurrent blocks in QRNNs, convolutional units in QCNNs and prediction heads in both architectures.

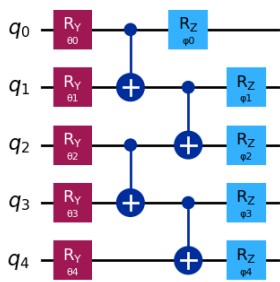

Figure 4: Basic PQC ansatz

## 4.3 FEATURE EXTRACTION

To map quantum states into features, we consider two strategies:

**(i) Sampling.** Projective measurement yields bitstrings $\{0,1\}^n$, which can be interpreted as token samples. While conceptually aligned with generative modeling, this method is difficult to train: optimization landscapes become noisy and semantically related tokens may map to distant bitstrings.

**(ii) Observable estimation.** Measuring expectation values of $Z$ and $ZZ$ operators on hidden registers produces continuous features for a classical linear projection. This yields smoother gradients, better robustness to shot noise, and greater semantic flexibility.

Although both methods rely on finite sampling in practice, we found observable-based features to be consistently more stable. We therefore use this approach throughout our experiments: we measure $Z$ on all *output* qubits and $ZZ$ on all pairs, yielding a feature vector of size $d + d(d-1)/2$ for $d$ qubits.

## 4.4 QUANTUM RECURRENT NEURAL NETWORK (QRNN)

Our QRNN architecture is built around two registers: an embedding register $\mathcal{E}$ and a hidden register $\mathcal{H}$. Each token is encoded into $\mathcal{E}$ with $R_y$ rotations, then transferred to $\mathcal{H}$ through a layer of CNOTs that establish correlations between the new input and the hidden state. The recurrent block $\mathcal{U}_{\text{rec}}$ applies parameterized $R_y$, $R_z$ rotations and entangling gates on $\mathcal{H}$, thereby updating the state across timesteps. The final hidden state is further processed by a separate PQC $\mathcal{U}_{\text{pred}}$ applied on $\mathcal{H}$, and then mapped into $Z$ and $ZZ$ expectation values for the classical projection layer.

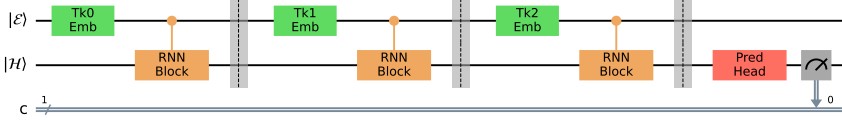

Figure 5: QRNN: tokens are embedded into $\mathcal{E}$, transferred to hidden register $\mathcal{H}$ by CNOTs, updated by recurrent PQC $\mathcal{U}_{\text{rec}}$, and passed through prediction PQC $\mathcal{U}_{\text{pred}}$ for observable-based feature extraction.

While inspired by prior work on quantum recurrent models (Bausch, 2020; Widdows et al., 2024a), our design was adapted specifically to the low-connectivity *heavy-hex* topology of IBM's Eagle and Heron processors. The placement of CNOT gates and grouping of rotations were chosen to minimize SWAP operations and circuit depth, ensuring better fidelity on real devices. Detailed circuit diagrams, including qubit layouts and hardware mappings, are provided in App. C.3.

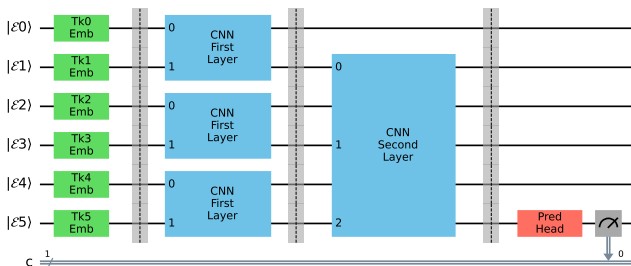

Figure 6: QCNN: tokens are embedded into registers $\mathcal{E}_0...\mathcal{E}_5$, processed by convolutional blocks $\mathcal{U}_{\text{conv}}$, and aggregated via prediction PQC $\mathcal{U}_{\text{pred}}$.

### 4.5 QUANTUM CONVOLUTIONAL NEURAL NETWORK (QCNN)

The QCNN variant follows the principle of local convolutions and pooling (Cong et al., 2019; Hur et al., 2022), but adapted to token-level sequence modeling. Tokens are embedded into parallel registers $\mathcal{E}_0, ..., \mathcal{E}_d$, which are grouped into overlapping neighborhoods and processed by convolutional blocks $\mathcal{U}_{\text{conv}}$. Each block processes 2 or 3 adjacent registers and consists of a 2-layered PQC as described in §4.2 to ensure better information flow. Pooling is implemented by only taking 1 register to the next layer, reducing the effective qubit count and yielding hierarchical feature representations. A prediction block $\mathcal{U}_{\text{pred}}$ processes the remaining register, whose state is finally mapped into $Z$ and $ZZ$ observables for classification by the projection head.

Compared to the sequential QRNN, the QCNN offers shallower depth and greater parallelism, making it attractive for NISQ devices. Our circuit designs were explicitly optimized for IBM hardware by aligning convolutional registers to heavy-hex connectivity and minimizing routing overhead. Full layer diagrams and hardware mapping strategies can be found in App. C.3.

### 4.6 OPTIMIZATION STRATEGY

Quantum parameters are trained with multi-sample *Simultaneous Perturbation Stochastic Approximation (SPSA)* (Spall, 1998), which estimates gradients by evaluating losses at pairs of symmetric points along random perturbation directions:

$$\hat{\nabla}_{\boldsymbol{\theta}} L \approx \frac{1}{P} \sum_{p=1}^{P} \frac{L(\boldsymbol{\theta} + \epsilon \boldsymbol{\delta}_p) - L(\boldsymbol{\theta} - \epsilon \boldsymbol{\delta}_p)}{2\epsilon} \, \boldsymbol{\delta}_p^{-1}. \tag{5}$$

This reduces circuit evaluations compared to parameter-shift or full finite differences while remaining hardware-friendly. The classical projection layer is trained with exact gradients via backpropagation.

## 5 EXPERIMENTS

We evaluate our proposed hybrid quantum language models on both synthetic datasets and established benchmarks from the quantum natural language processing (QNLP) literature.

### 5.1 DATASETS

**Literature benchmarks.** To enable comparison with prior work, we also evaluate on established QNLP classification datasets (Lorenz et al., 2023):

- **MC (Meaning Classification)**: Binary classification with 70 train and 30 test 4-word sentences in two classes: *programming* and *cooking*. We also derive a language modeling version (**MC-LM**) using the same sentences.
- **RP (Relative Pronoun resolution)**: Binary classification with 74 train and 31 test 4-word sentences, grouped by sentence structure: *X that did Y* vs. *X that Y did*.

**Synthetic language modeling dataset.** To evaluate the ability of our models to capture compositional structure in natural language, we also generate a small-scale **Toy Sentence Language Modeling (TS-LM)** dataset. The vocabulary contains 24 unique words,

grouped into categories such as *subjects*, *verbs*, *adjectives*, *objects*, *prepositions*, and *locations*. Sentences are sampled from a context-free grammar ensuring basic syntactic consistency: `subject-verb-[adjective]-object-[preposition-location]`. For example, a sentence may look like: "`man sees small dog on table`". The dataset consists of 200 training and 50 test sentences, with sentence lengths between 3 and 6 words. This controlled setup balances variability with tractability for small-scale quantum models. The dataset and generation code are available in the supplementary material.

## 5.2 EXPERIMENTAL SETUP

**Models and Baselines.** We evaluate:

- **Hybrid QLMs (ours)**: QRNN and QCNN architectures (§4), trained end-to-end with SPSA for quantum parameters and backpropagation for the classical projection head.
- **Classical baselines**: FFNN, RNN, LSTM, CNN, and Transformer models with comparable parameter counts.

**Training Setup.** All models are trained with Adam. Gradients for the quantum parameters are estimated using multi-sample SPSA, while the classical projection head is trained with exact gradients. Most experiments are performed on simulators; for QRNN and QCNN, we additionally evaluate trained models on real IBM hardware (Eagle/Heron processors). For MC and TS-LM, we also train directly on hardware. Embedding registers use `emb_size=3` qubits. We tune learning rate, batch size, and epochs per task. SPSA uses population size $p = 8$ and perturbation scale $\sigma = 0.05$. Full hyperparameters are given in App. C.1. Detailed information about circuit complexity, number of trainable parameters, and per-epoch training costs for each architecture is provided in App. C.2.

**Evaluation Protocol.** For **language modeling** tasks, we report train/test perplexity (Tr PPL / Ts PPL) and the average probability assigned to the correct next token in the test set (Acc.). For **binary classification** tasks, we report test accuracy.

## 5.3 MAIN RESULTS

Table 1 summarizes the performance of classical and quantum models. Classical baselines achieve strong results across tasks, with RNNs, LSTMs, and Transformers performing best overall.

For quantum models, both QRNN and QCNN achieve accuracies and perplexities comparable to their classical counterparts in simulation. On MC-LM and TS-LM, quantum models reach test perplexities in the same range as classical networks, showing that relatively shallow PQC-based architectures can capture sequential patterns in small-scale language modeling. On the MC classification task, QRNN and QCNN match the perfect accuracy of classical models, while on RP they achieve slightly lower but still competitive accuracies (80.6% and 83.9% vs. 90.3% for RNN). Note however that the RP test set of 31 samples contains 20 words not seen during training, and 4 of the 31 samples cannot be inferred by any model from the train set alone due to inherent ambiguity. Thus, the maximum theoretical accuracy without taking into acount random guesses is 87.1%, and the QRNN and QCNN achieve scores close to this limit.

Hardware runs highlight the gap between ideal simulation and current devices. Models trained on simulators but evaluated on hardware (**Real Hardware Eval**) show moderate accuracy/perplexity degradation due to noise which decreases with more circuit samples and better hardware: for example the 5.83 test perplexity of QCNN on MC-LM was obtained with 10k shots on a IBM Heron processor, while lowering the shot count or running on previous generation Eagle processors yields significantly higher perplexities. Fully hardware-trained models (**Real Hardware Train + Eval**) perform worse still, reflecting harder optimization due to noisy sampling. Notably, QRNNs are somewhat more robust than QCNNs on hardware, likely due to their lower qubit counts and fewer parameters.

Overall, these results indicate that (i) Hybrid quantum language models can match the performance of small classical models on toy NLP benchmarks in simulation. (ii) Noise remains the primary bottleneck for real devices, and (iii) With the development of larger quantum devices, we can move beyond very simple QNLP models such as DisCoCat, which were limited to just a handful of qubits. Our results show that more expressive architectures like QRNNs and QCNNs can be successfully

Table 1: Performance of classical and quantum language models across multiple tasks. Columns MC and RP report binary classification accuracy (%), while MC-LM and TS-LM denote two next-token prediction tasks with training/test perplexity and average next token prediction accuracy. For QRNN and QCNN, we report: (i) **Simulator** results with noiseless statevector simulation; (ii) **Real Hardware Eval**, where the simulator-trained parameters are executed on quantum processors; and (iii) **Real Hardware Train + Eval**, where both training and evaluation are carried out directly on quantum hardware.

| Training Setup | Model | Test Acc. [%] | | MC-LM | | | TS-LM | | |
| | | MC | RP | Tr PPL | Ts PPL | Acc. [%] | Tr PPL | Ts PPL | Acc. [%] |
|---|---|---|---|---|---|---|---|---|---|
| **Classical** Autograd Backprop | FFNN | 100 | 93.5 | 3.79 | 6.30 | 20.0 | 3.38 | 3.51 | 31.0 |
| | RNN | 100 | 90.3 | 4.12 | 5.44 | 20.0 | 3.52 | 3.75 | 28.7 |
| | LSTM | 100 | 96.8 | 4.01 | 5.79 | 19.0 | 3.82 | 3.89 | 27.4 |
| | CNN | 100 | 80.6 | 3.80 | 5.07 | 22.0 | 3.73 | 3.65 | 30.8 |
| | Transf | 100 | 83.9 | 3.81 | 4.81 | 23.0 | 3.39 | 3.47 | 32.0 |
| **Quantum** Simulator | QRNN | 100 | 80.6 | 4.64 | 4.84 | 22.6 | 3.66 | 3.47 | 31.6 |
| | QCNN | 100 | 83.9 | 5.10 | 5.69 | 19.0 | 3.96 | 3.76 | 28.9 |
| | DisCoCat | 79.8 | 72.3 | / | / | / | / | / | / |
| Real Hardware Eval | QRNN | 100 | 74.2 | / | 4.86 | 22.4 | / | 3.86 | 28.4 |
| | QCNN | 100 | 77.4 | / | 5.83 | 18.4 | / | 4.43 | 25.0 |
| Real Hardware Train + Eval | QRNN | 100 | / | / | / | / | 4.60 | 4.82 | 24.8 |
| | QCNN | 100 | / | / | / | / | 8.82 | 8.65 | 12.4 |
| | DisCoCat | 83.3 | 67.7 | / | / | / | / | / | / |

trained on today's hardware. However, careful adaptation of circuit design to the hardware topology (§4) remains crucial to achieve robust performance.

## 5.4 ABLATION STUDIES

**Training Randomness**  In Table 2, we study the effect of training randomness by reporting the mean and standard deviation of test perplexity or accuracy over 5 independent runs with different random seeds for each of the considered tasks on both QRNN and QCNN architectures. We observe that both models exhibit rather high variance across runs, indicating sensitivity to initialization and stochasticity in the training process. The effect is more pronounced for QCNN, which has higher qubit and quantum parameter counts, leading to a higher dimensionality of the optimization landscape and therefore more challenges in finding optimal solutions, including very weak gradient signal in barren plateaus. This suggests that further work is needed to improve training stability, potentially through better initialization schemes, regularization techniques, or more robust optimization methods.

Table 2: Effect of training randomness. We report the mean and standard deviation of test perplexity or accuracy over 5 independent runs with different random seeds for each of the considered tasks on both QRNN and QCNN architectures.

| Task | Model | Avg Score | Stdev |
|---|---|---|---|
| MC (Acc) | QRNN | 100 | 0.0 |
| | QCNN* | 85.3 | 19.1 |
| RP (Acc) | QRNN | 69.6 | 7.1 |
| | QCNN | 78.7 | 3.7 |
| MC-LM (PPL) | QRNN | 5.00 | 0.19 |
| | QCNN** | 6.28 | 0.62 |
| TS-LM (PPL) | QRNN | 4.12 | 0.62 |
| | QCNN | 4.28 | 0.62 |

\* 3 of 5 runs have scores of 100%, others fail to learn
\*\* 1 of 5 runs had a PPL of 13.3 so we consider it an outlier

**Number of shots**  In Figure 7a, we show how the number of shots used for expectation estimation affects QRNN training. Increasing shots generally speeds up training by providing more accurate loss estimates, with diminishing returns beyond 4096 shots. Even 256 shots allow effective learning, demonstrating robustness to shot noise. Training on real hardware with 256 shots yields comparable performance to simulations, highlighting that near-term devices can provide useful gradient information despite noise.

**Embedding Size**  In Figure 7b, we illustrate the effect of embedding size (number of qubits per embedding register) on QRNN training. Larger embeddings reduce training loss by enabling more expressive representations, but gains saturate beyond a certain point, indicating an optimal

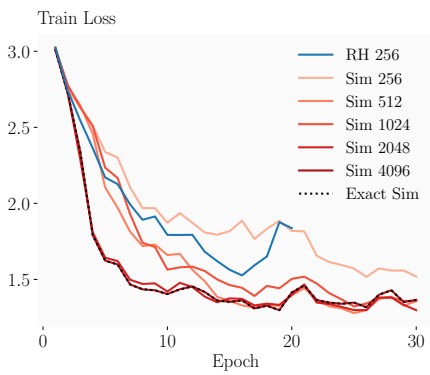

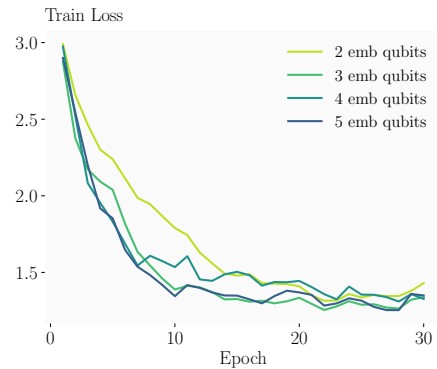

(a) Train Loss evolution for different shot counts used for estimation in both Simulator and Real Hardware.

(b) Train Loss evolution for different embedding sizes (number of qubits).

Figure 7: Train Loss evolution for different hyperparameters when training QRNN models on TS-LM.

qubit count. Note that the total qubit count includes both embedding and hidden registers: for example, 2 embedding qubits $\rightarrow$ 4 total, $5 \rightarrow 12$ total, and $8 \rightarrow 19$ total. While larger embeddings improve expressivity, they also increase the number of quantum parameters and the complexity of the optimization landscape, making training more challenging. On simulators, larger embeddings increase computational cost exponentially (20 s/epoch, 65 s/epoch, 1150 s/epoch for the three examples), whereas on real hardware timing remains roughly constant ($\approx$1000 s/epoch) due to communication overheads. This highlights the trade-off between expressivity, trainability, and practical efficiency when choosing embedding size.

## 6 LIMITATIONS

Training and evaluation on real quantum hardware highlight several limitations. First, noise and finite-shot effects degrade performance, with a noticeable gap between simulator and hardware results. QRNNs tend to be more robust than QCNNs on hardware, likely due to lower qubit counts and fewer variational parameters. Second, models are sensitive to initialization and stochasticity, especially QCNNs with larger circuits, which exhibit higher variance across runs (Table 2). Third, scaling the embedding size improves expressivity but comes at the cost of exponentially higher simulation time and increased circuit complexity; careful selection of the number of qubits is therefore critical. Finally, while our circuits outperform simple models such as DisCoCat, achieving better results requires both increased hardware resources and careful adaptation of circuit design to the low-connectivity, heavy-hex lattice of modern IBM devices (§4).

These limitations emphasize that, although current HQLMs are promising for near-term quantum NLP, practical deployment on larger and more realistic datasets will require improvements in hardware noise mitigation, circuit optimization, and training stability.

## 7 CONCLUSIONS

We introduced hybrid quantum language models for sequence modeling, combining PQC-based QRNN and QCNN architectures with classical projection heads. Our results show that these models can match small classical networks in simulation and demonstrate the feasibility of training and evaluating quantum sequence models on current NISQ devices.

Key insights include: (i) estimator-based feature extraction provides smoother gradients and stabilizes training, (ii) model performance scales with embedding size up to an optimal qubit count, balancing expressivity and computational cost, and (iii) careful circuit design adapted to hardware topology is essential for robustness on real devices.

Our work establishes a foundation for more expressive quantum NLP models and highlights the trade-offs between model complexity, hardware constraints, and training stability. Future directions include scaling to larger datasets, incorporating noise-resilient circuit designs, and exploring hybrid optimization strategies to improve training robustness on real quantum hardware.

## REPRODUCIBILITY STATEMENT

We release the complete code used for our experiments at [ANONYMISED FOR REVIEW] - available in Supplementary Materials. A detailed description of the experimental setup and hyperparameters is provided in App. C.

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

## A    LLM USAGE DISCLAIMER

This paper was prepared using the following Large Language Models (LLMs): GPT 4.1 and GPT-5 (OpenAI). The authors have reviewed and edited the content produced by these models, and take full responsibility for the final content of the publication. The LLMs were used to assist with drafting and editing text, improving grammar and style, and suggesting rephrasings. No scientific claims or results were generated by the LLMs; all technical content, experiments, and conclusions are the original work of the authors. The use of LLMs was limited to non-technical writing tasks to enhance clarity and readability. Additionally, the LLMs were used for searching literature for relevant papers and resources. The authors acknowledge the potential limitations and biases of LLMs, and have carefully verified all information in the final manuscript. Any errors or inaccuracies are solely the responsibility of the authors.

## B    NOTATION

### B.1    QUANTUM REGISTERS AND PARAMETER NOTATION

In our hybrid quantum language models, we explicitly distinguish between different quantum registers, each with its own purpose and number of qubits. Specifically, we consider:

- **Embedding register** $\mathcal{E}$ with $d_e$ qubits, which encodes the input token embeddings.

- **Hidden register** $\mathcal{H}$ with $d_h$ qubits, which stores recurrent or latent states.

- **Output or prediction register** $\mathcal{O}$ with $d_o$ qubits, optionally used for feature extraction or measurement.

Each register has its own Hilbert space, and the total system is represented as

$$\mathcal{H}_{\text{total}} = \mathcal{H}_{\mathcal{E}} \otimes \mathcal{H}_{\mathcal{H}} \otimes \mathcal{H}_{\mathcal{O}} \cong \mathbb{C}^{2^{d_e + d_h + d_o}}. \tag{6}$$

Individual qubits within a register are indexed by lowercase letters, e.g., $q_j \in \mathcal{E}$ or $h_k \in \mathcal{H}$. A tensor-product quantum state can be written as

$$|\psi_{\text{in}}\rangle = \bigotimes_{j=1}^{d_e} R_y(\theta_j) |0\rangle_{q_j} \otimes \bigotimes_{k=1}^{d_h} |0\rangle_{h_k}. \tag{7}$$

To simplify notation, we often write the all-zero state of $d$ qubits as $|\mathbf{0}_d\rangle$, or simply $|\mathbf{0}\rangle$ when the dimension is clear from context.

**Parameter vectors.**    Token embeddings are represented by trainable vectors $\boldsymbol{\theta}_v \in \mathbb{R}^{d_e}$ for each token $v \in V$, which are mapped to the embedding register via $R_y$ rotations. Hidden registers and entangling layers are parametrized separately, e.g., $\boldsymbol{\phi} \in \mathbb{R}^{n_h}$. For convenience, all trainable quantum parameters can be concatenated into a master vector $\boldsymbol{\Theta} = [\boldsymbol{\theta}, \boldsymbol{\phi}]$ when describing gradient updates or optimization procedures.

**Observables and measurements.**    When measuring quantum states, we explicitly specify the register of interest. For example, the $Z$ observable over the hidden register is denoted

$$\hat{Z}_{\mathcal{H}} = \sum_{j \in \mathcal{H}} Z_j, \tag{8}$$

which is used to extract features for the classical output layer. This notation clarifies which registers contribute to the model output, particularly in hybrid architectures with multiple quantum sub-registers.

### B.2 QUANTUM GATES AND CIRCUIT OPERATIONS

Quantum circuits manipulate qubits via unitary gates. Single-qubit rotations, particularly $R_y$ gates, are used to encode token embeddings:

$$R_y(\theta_j) |0\rangle_{q_j} = \cos(\theta_j/2) |0\rangle_{q_j} + \sin(\theta_j/2) |1\rangle_{q_j}, \quad q_j \in \mathcal{E}. \tag{9}$$

Multi-qubit interactions are introduced through entangling gates, such as controlled-NOT (CNOT) operations:

$$\text{CNOT}_{c,t} |q_c q_t\rangle = \begin{cases} |q_c q_t \oplus 1\rangle, & \text{if } q_c = 1, \\ |q_c q_t\rangle, & \text{otherwise,} \end{cases} \tag{10}$$

where $c$ and $t$ denote control and target qubits, respectively. Entangling gates are primarily applied between embedding and hidden registers, or within the hidden register, to capture correlations across tokens.

We denote the full parametrized circuit acting on registers $\mathcal{E}$ and $\mathcal{H}$ as

$$U(\boldsymbol{\Theta}) = \prod_{l=1}^{L} U_l(\boldsymbol{\Theta}_l), \tag{11}$$

where each layer $U_l$ can contain both single-qubit rotations and entangling operations, and $\boldsymbol{\Theta}_l \subset \boldsymbol{\Theta}$ represents the parameters in that layer. The resulting quantum state is then

$$|\psi(\boldsymbol{\Theta})\rangle = U(\boldsymbol{\Theta}) |\mathbf{0}\rangle, \tag{12}$$

with $|\mathbf{0}\rangle$ the all-zero state of the entire system, as defined previously.

**Observables.** Measurement operators are associated with specific registers to extract features for the classical output layer. For example, for the hidden register $\mathcal{H}$, $Z$ and $ZZ$ operators are used to compute expectation values:

$$f(\boldsymbol{\Theta}) = \langle \psi(\boldsymbol{\Theta}) | \hat{O}_{\mathcal{H}} | \psi(\boldsymbol{\Theta}) \rangle, \tag{13}$$

where $\hat{O}_{\mathcal{H}}$ denotes the collection of observables applied to $\mathcal{H}$. These expectation values serve as inputs to classical layers, providing a hybrid quantum-classical representation.

## C EXPERIMENTAL DETAILS

### C.1 HYPERPARAMETERS AND TRAINING DETAILS

Detailed hyperparameters for all experiments are summarized in Table 3. We tune learning rate, batch size, and number of epochs per task. SPSA uses population size $p = 8$ and perturbation scale $\sigma = 0.05$.

Table 3: Hyperparameters for all experiments.

| Task | Model | Seq Len | CNN kernels | Learning Rate | Batch Size | Epochs |
|------|-------|---------|-------------|---------------|------------|--------|
| MC | QRNN | 4 | / | 0.1 | 10 | 20 |
| MC | QCNN | 6 | 3,3 | 0.1 | 10 | 20 |
| RP | QRNN | 4 | / | 0.1 | 10 | 40 |
| RP | QCNN | 4 | 2,2 | 0.1 | 10 | 40 |
| MC-LM | QRNN | 4 | / | 0.1 | 16 | 40 |
| MC-LM | QCNN | 6 | 3,3 | 0.1 | 16 | 40 |
| TS-LM | QRNN | 6 | / | 0.1 | 32 | 30 |
| TS-LM | QCNN | 6 | 3,3 | 0.1 | 32 | 30 |

### C.2 TRAINING DETAILS

Detailed information about circuit complexity, number of trainable parameters, and per-epoch training costs for each architecture is provided in Table 4.

Table 4: Model and training details for all architectures on the TS-LM task. Time per epoch is measured on an AMD Ryzen AI 9HX with 32GB RAM for classical models and simulators, and IBM Eagle/Heron processor for quantum models (in parentheses, the actual quantum processing time excluding overhead). For quantum models, we report two embedding sizes (E=3,10 for QRNN, E=3,6 for QCNN). For QRNN E=10 we have 17 hidden qubits, but we limit the $ZZ$ observables to pairs of adjacent qubits only, reducing the final feature size to 19.

| Type | Model | Params Total (Q+C) | Quantum Circ | | | | Time per epoch clock time (Q usage) |
| | | | Qubits | Total gates | 2Q gates | 2Q depth | |
|---|---|---|---|---|---|---|---|
| Classical | FFNN | 316 (0+316) | | | | | <1s |
| | RNN | 256 (0+256) | | | | | <1s |
| | LSTM | 376 (0+376) | | | | | <1s |
| | CNN | 304 (0+304) | | | | | <1s |
| | Transf | 384 (0+384) | | | | | <1s |
| Quantum Simulator | QRNN (E=3) | 258 (90+168) | 6 | 106 | 34 | 22 | 20s |
| | QCNN (E=3) | 316 (148+168) | 19 | 181 | 48 | 12 | 110s |
| | QRNN (E=10) | 1158 (342+816) | 27 | 1966 | 188 | 22 | >2h |
| | QCNN (E=6) | 1044 (340+704) | 37 | 1213 | 102 | 12 | OOM |
| Quantum Hardware | QRNN (E=3) | 258 (90+168) | 6 | 370 | 34 | 22 | 16min (7min) |
| | QCNN (E=3) | 316 (148+168) | 19 | 577 | 48 | 12 | 35min (7min) |
| | QRNN (E=10) | 1158 (342+816) | 27 | 1966 | 188 | 22 | 40min (8min) |
| | QCNN (E=6) | 1044 (340+704) | 37 | 1213 | 102 | 12 | 80min (8min) |

## C.3 CIRCUIT DETAILS

Figure 8 and Figure 10 show detailed circuit diagrams for QRNN and QCNN architectures, respectively. Example qubit layouts on IBM Heron processor for different embedding sizes are shown in Figure 9 and Figure 11.

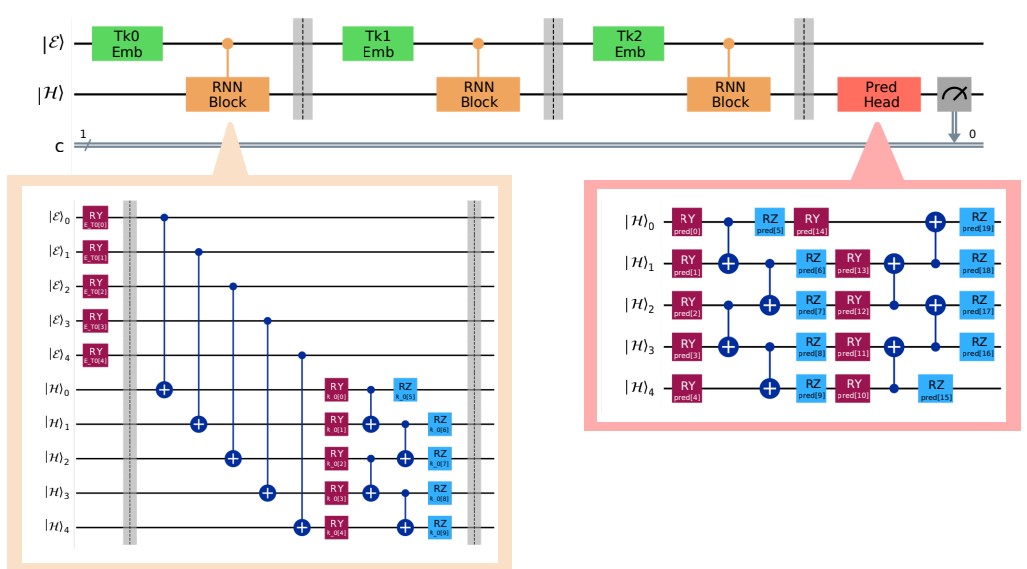

Figure 8: QRNN: tokens are embedded into $E$, transferred to hidden register $H$ by CNOTs, updated by recurrent PQC $\mathcal{U}_{\mathrm{rec}}$, and passed through prediction PQC $\mathcal{U}_{\mathrm{pred}}$ for observable-based feature extraction.

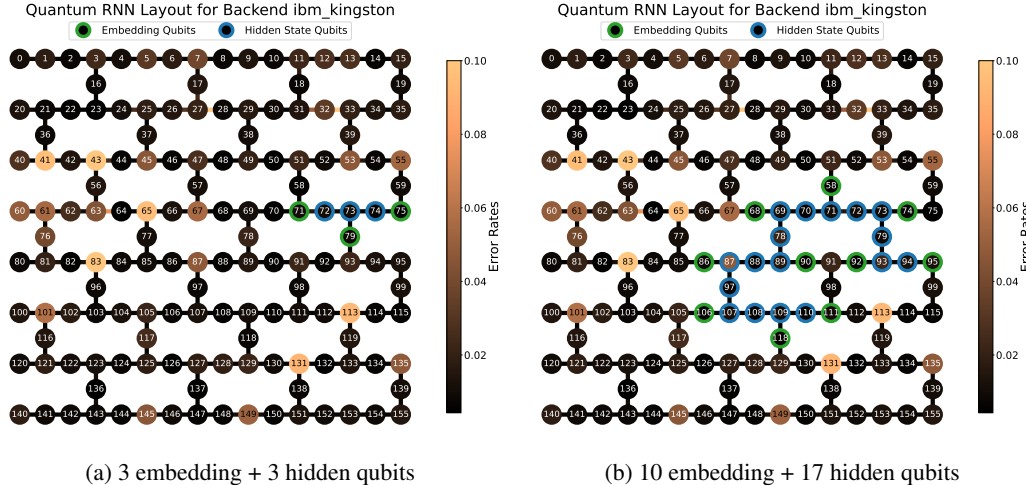

(a) 3 embedding + 3 hidden qubits        (b) 10 embedding + 17 hidden qubits

Figure 9: Example qubit layout on IBM Heron processor for QRNN with (a) 3-qubit embedding and (b) 10-qubit embedding. The heavy-hex connectivity is highlighted, along with error rates for single and 2 qubit gates. **Green** qubits are used for embedding register $\mathcal{E}$ and are not connected. **Blue** qubits are used for hidden register $\mathcal{H}$ and need to be connected. Some of the hidden qubits are auxiliary and do not correspond to embedding qubits in order to respect the hardware connectivity.

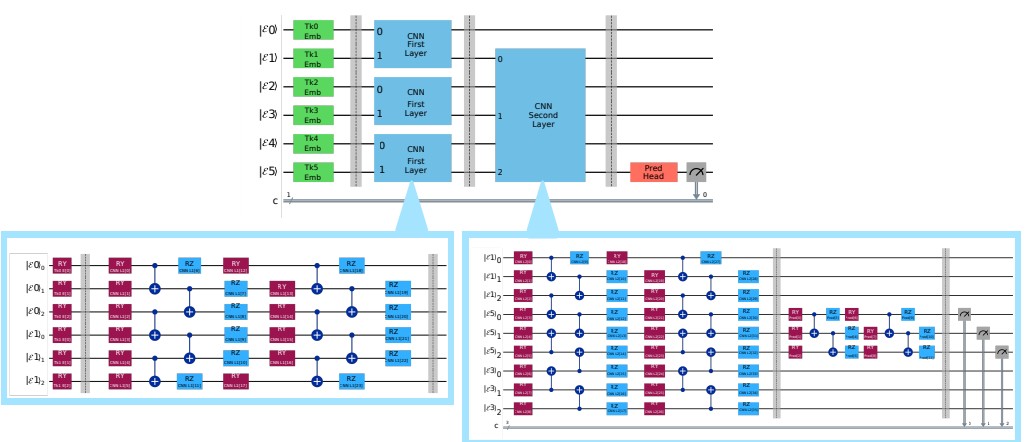

Figure 10: QCNN: tokens are embedded into registers $\mathcal{E}_0...\mathcal{E}_5$ then processed by convolutional blocks $\mathcal{U}_{\text{conv1}}$ and $\mathcal{U}_{\text{conv2}}$, and prediction PQC $\mathcal{U}_{\text{pred}}$.

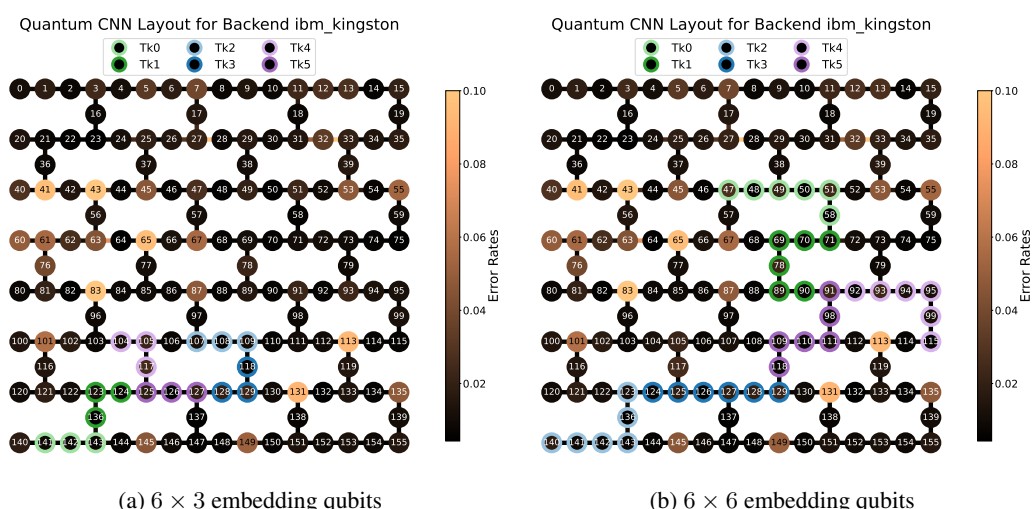

(a) $6 \times 3$ embedding qubits                    (b) $6 \times 6$ embedding qubits

Figure 11: Example qubit layout on IBM Heron processor for QCNN with (a) $6 \times 3$ embedding and (b) $6 \times 6$ embedding. The heavy-hex connectivity is highlighted, along with error rates for single and 2 qubit gates. Each color represents one embedding register $\mathcal{E}_i$, where qubits need to be connected. The two shades of each color represent the connections made by the first convolutional block. The darker shade of each color represents the qubits used in the second convolutional block. **Dark Purple** qubits are used for measurements as prediction register $\mathcal{O}$.

