# OpenReview forum: "Practical Hybrid Quantum Language Models with Observable Readout on Real Hardware"
_ICLR.cc/2026/Conference — Submitted to ICLR 2026_

### Official Review · Reviewer_1pve · 2025-10-14

**Soundness:** 3
**Presentation:** 3
**Contribution:** 2
**Rating:** 4
**Confidence:** 4

**Summary:**

The paper builds on the growing area of hybrid quantum-classical language modeling. It introduces a framework that uses observable-based readouts from parameterized quantum circuits to process sequential data, combining quantum feature extraction with classical post-processing. The approach is designed to run on real hardware, and the authors present results from small-scale experiments showing that the model can be trained end-to-end with noisy measurements.

**Strengths:**

In my view, this work includes practical considerations for training stability, circuit depth, and noise handling, which makes it one of the few studies that attempt a full pipeline rather than just simulations.

The paper is clearly written and does a good job explaining its hybrid architecture and training setup. It takes the extra step of running experiments on real quantum hardware, which adds some practical credibility.

The results, though small in scale, are consistent and show that end-to-end training is at least feasible on current devices. I think that is sufficient given the state of current hardware.

**Weaknesses:**

The paper’s main idea, using observable-based quantum readouts for sequence modeling, is only a small step forward rather than a clear new direction. Similar hybrid quantum–classical setups have already been explored in fine-tuning and compositional NLP studies such as arXiv:2504.08732 and arXiv:2409.08777. The framing of using quantum circuits for text-like data tasks is therefore not particularly new, and the contribution is mostly in applying existing ideas to a slightly different problem setting.

The focus on observable-based readout is practical but limited in scope. It simplifies measurement and training but does not offer clear theoretical or empirical benefits over standard hybrid models that use expectation-based features or variational encoders. Also, in my view, the claimed advantage of better noise handling or stability is not supported by strong comparative results.

The use of real hardware is presented as a key achievement, but the experiments are too small to show meaningful performance or scalability. Running shallow circuits on short sequences mainly demonstrates feasibility, not effectiveness. The work stops short of showing that the observable-based design can scale to larger datasets, longer sequences, or more expressive models.

Overall, the paper’s contribution feels modest. It applies known methods in a narrow setup, emphasizes hardware execution without deeper insight, and lacks convincing evidence that observable-based readouts offer any real progress in hybrid quantum language modeling.

**Questions:**

What is the novelty compared to prior quantum language processing works?

---

### Official Review · Reviewer_YGyU · 2025-10-28

**Soundness:** 2
**Presentation:** 2
**Contribution:** 1
**Rating:** 0
**Confidence:** 4

**Summary:**

The authors propose a model that combines quantum and classical neural networks. They compare its performance with purely classical models having similar parameter counts. The hybrid model is tested on real quantum hardware.

**Strengths:**

This work includes results from real quantum hardware.

**Weaknesses:**

1. All elements (QRNN, QCNN, QNLP, SPSA, etc.) are well known. The results show no surprisingly good performance and only match classical baselines with similar parameter counts (a few hundred). This limits the work's novelty and contribution.
2. The task is limited to binary classification and text generation with a synthetic small dataset, raising questions about real-world applicability.
3. The paper lacks theoretical guarantees and discussion of the approach's foundations.

**Questions:**

1. How can we be sure the contribution comes from the quantum part rather than the classical projection part? In Table 4, some of the classical components in the hybrid quantum-classical setting already have more parameters than the classical baselines.
2. Why choose hybrid quantum models over classical models? What advantage do they offer?

---

### Official Review · Reviewer_huH1 · 2025-10-30

**Soundness:** 2
**Presentation:** 3
**Contribution:** 2
**Rating:** 2
**Confidence:** 3

**Summary:**

This paper proposes hybrid quantum–classical architectures for sequence modeling, focusing on a quantum recurrent neural network (QRNN) and a quantum convolutional neural network (QCNN). These models use shallow parameterized quantum circuits combined with a lightweight classical projection layer. The authors train quantum parameters with a stochastic perturbation method (SPSA) and classical parameters with gradient descent. The main claim is that the proposed models can be trained and evaluated end-to-end on current IBM quantum hardware. Experiments on small synthetic language modeling tasks and reasoning puzzles show that the models can roughly match small classical baselines.

**Strengths:**

* The paper represents a solid engineering effort to implement quantum sequence models on real hardware. The circuits are designed with attention to connectivity and noise limitations, and the authors report hardware-specific details such as gate counts, layouts, and shot configurations.
* The manuscript is clearly written, with good structure and detailed appendices. Figures and tables are informative, and experimental settings are transparent.
* The work includes ablations on number of shots, embedding size, and training variance, which provide useful practical insights into how these hybrid models behave under noise.

**Weaknesses:**

* The paper does not convincingly explain why quantum circuits are needed for language modeling. It suggests that quantum computation might provide richer representations but offers no theoretical or empirical justification. There is no discussion of what properties of language data could benefit from quantum operations or what specific limitation of classical sequence models this work intends to overcome.
* The findings primarily confirm that current hardware can execute small parameterized circuits, not that quantum models provide new capabilities for language tasks.
*  The datasets contain only a few hundred samples and simple grammatical or logical patterns. These settings are too limited to reveal any generalization benefit or inductive bias. The conclusions drawn from them are therefore narrow.

**Questions:**

1. What concrete modeling limitation of classical RNNs or CNNs does this quantum design aim to solve?
2. Is there a coherent recurrent memory, is measurement performed at each step of the QRNN (unclear from the paper and diagrams)?
3. How does 2. above relate to SPSA to estimate gradients, which requires repeated circuit evaluations and measurement-based expectation estimation after each relevant step? And how does this relate to the actual device used?
4. Where is the nonlinearity?

---

### Official Review · Reviewer_6DAV · 2025-11-05

**Soundness:** 3
**Presentation:** 3
**Contribution:** 3
**Rating:** 4
**Confidence:** 3

**Summary:**

This paper introduces Hybrid Quantum Language Models (HQLMs) that integrate quantum recurrent neural networks (QRNNs) and quantum convolutional neural networks (QCNNs) with a classical projection layer. It presents what appears to be the first demonstration of generative language modeling trained and evaluated on real quantum hardware, moving beyond simulation-based studies in quantum NLP.

The study leverages parametric quantum circuits (PQCs) trained via a multi-sample Simultaneous Perturbation Stochastic Approximation (SPSA) method for quantum parameters and gradient-based optimization for classical layers. Experiments are conducted both on simulators and real IBM NISQ devices (Eagle and Heron processors).

**Strengths:**

- Introduction of QRNN and QCNN as hardware-feasible quantum analogues of classical sequence models.

- Use of multi-sample SPSA for efficient and hardware-compatible gradient estimation, combined with standard backpropagation on classical layers.

- Synthetic dataset (TS-LM) for next-word prediction, code and circuits made available for replication.

**Weaknesses:**

- The benchmarked dataset migth be too toy for NLP
- It is unclear whether the models are scalable
- It might help to discuss the relation/connections with existing work like  https://arxiv.org/pdf/2302.13812

**Questions:**

- In which sense is the training framework scalable, as mentioned in the contribution section?

- Could you show the cons and prons of QCNN and QRNN?  It may help to clarify the relative strengths of QCNNs and QRNNs, identifying scenarios where QCNNs demonstrate superior performance and, conversely, situations in which QRNNs provide greater benefits.

- Does QCNN/QRNN benefits from scaling, e.g. data scale ande model scale? if no, why?

- Why don't you try a quantum transformer? What are the challenges to build quantum transformer?

---

### Meta-Review · Area_Chair_NcrV · 2025-12-24

**Summary:**

This paper proposed to study hybrid quantum language models (QRNNs and QCNNs). However, as the reviewers pointed out, the paper has the following issues:

- The benchmarked dataset migth be too toy for NLP.
- It is unclear whether the models are scalable.
- The paper does not convincingly explain why quantum circuits are needed for language modeling.

The scores are unanimously rejects. Considering all, the final decision is rejection.

**Reviewer Concerns:**

The authors didn't attend rebuttal, and there are notable outstanding issues raised by all reviews.

**Reviewer Scores:**

I don't think the scores will be changed - the authors didn't involve into the rebuttal and the scores are unanimously rejects.

---

### Decision · Program_Chairs · 2026-01-26

Reject